# Preliminary Structure–Activity Relationship Study of the MMV Pathogen Box Compound MMV675968 (2,4-Diaminoquinazoline) Unveils Novel Inhibitors of *Trypanosoma brucei brucei*

**DOI:** 10.3390/molecules27196574

**Published:** 2022-10-04

**Authors:** Darline Dize, Rolland Bantar Tata, Rodrigue Keumoe, Rufin Marie Kouipou Toghueo, Mariscal Brice Tchatat, Cyrille Ngansop Njanpa, Vianey Claire Tchuenguia, Lauve Tchokouaha Yamthe, Patrick Valere Tsouh Fokou, Benoît Laleu, James Duffy, Ozlem Tastan Bishop, Fabrice Fekam Boyom

**Affiliations:** 1Antimicrobial and Biocontrol Agents Unit (AmBcAU), Laboratory for Phytobiochemistry and Medicinal Plants Studies, Department of Biochemistry, Faculty of Science, University of Yaoundé I, Yaoundé P.O. Box 812, Cameroon; 2Research Unit in Bioinformatics (RUBi), Department of Biochemistry and Microbiology, Rhodes University, Grahamstown 6139, South Africa; 3Medicines for Malaria Venture, Route de Pré-Bois 20, 1215 Meyrin, Switzerland

**Keywords:** *Trypanosoma brucei brucei*, MMV Pathogen Box, antitrypanosomal, MMV675968 (2,4-diaminoquinazoline), in silico, structure–activity relationship, DHFR inhibitor, time-kill kinetic, DNA fragmentation

## Abstract

New drugs are urgently needed for the treatment of human African trypanosomiasis (HAT). In line with our quest for novel inhibitors of trypanosomes, a small library of analogs of the antitrypanosomal hit (MMV675968) available at MMV as solid materials was screened for antitrypanosomal activity. In silico exploration of two potent antitrypanosomal structural analogs (**7**-MMV1578647 and **10**-MMV1578445) as inhibitors of dihydrofolate reductase (DHFR) was achieved, together with elucidation of other antitrypanosomal modes of action. In addition, they were assessed in vitro for tentative inhibition of DHFR in a crude trypanosome extract. Their ADMET properties were also predicted using dedicated software. Overall, the two diaminoquinazoline analogs displayed approximately 40-fold and 60-fold more potency and selectivity in vitro than the parent hit, respectively (MMV1578445 (**10**): IC_50_ = 0.045 µM, SI = 1737; MMV1578467 (**7**): IC_50_ = 0.06 µM; SI = 412). Analogs **7** and **10** were also strong binders of the DHFR enzyme in silico, in all their accessible protonation states, and interacted with key DHFR ligand recognition residues Val32, Asp54, and Ile160. They also exhibited significant activity against trypanosome protein isolate. MMV1578445 (**10**) portrayed fast and irreversible trypanosome growth arrest between 4–72 h at IC_99_. Analogs **7** and **10** induced in vitro ferric iron reduction and DNA fragmentation or apoptosis induction, respectively. The two potent analogs endowed with predicted suitable physicochemical and ADMET properties are good candidates for further deciphering their potential as starting points for new drug development for HAT.

## 1. Introduction

African trypanosomiasis is a neglected tropical disease with massive social and economic impacts in endemic regions, threatening both humans (Human African Trypanosomiasis, HAT) and animals (Animal African Trypanosomiasis, AAT) health [1,2]. The Human African Trypanosomiasis is distributed in 36 countries in sub-Saharan Africa with an estimated at-risk population of about 55 million between 2016 and 2020 [3] and is fatal unless treated. Although the disease is not completely eradicated, the multiple control strategies put in place have led to an important decline in the number of cases from almost 40,000 in 1998 to 992 and 663 new cases reported in 2019 and 2020, respectively [3]. However, these statistics are likely underestimated given that the majority of patients are in rural areas and war zones. On the other hand, the animal African trypanosomiasis remains one of the most important cattle diseases in sub-Saharan Africa [4]. Every year, about 3 million deaths and an estimated economic loss in cattle production in the range of USD 1.0–1.2 billion are attributed to this disease [5]. The management of African trypanosomiasis mostly relies on early diagnosis to better the prospect of an adequate treatment using available chemotherapeutic agents [3]. Despite progress, the current drugs available for treating HAT are inadequate, due to toxicity, poor efficacy and drug resistance [4,6]. In addition, the current treatments are inappropriate for a rural setting with poor facilities as they almost all require parenteral administration. The only oral option for both stage 1 and stage 2 HAT is fexinidazole, which was recently developed by the Drugs for Neglected Disease initiative (DNDi). There is therefore an urgent need for new treatments bearing novel modes of action for HAT, for the reasons given above, but also with the aim of elimination and eradication of this disease [3]. The new drugs should ideally be through oral regimens and have efficacy against both stages 1 and 2 of the disease. Beyond the scope of existing drugs in the clinic, there are several trypanosomatid-specific targets that have been extensively investigated in trypanosomiasis drug-discovery programs. Examples include several enzymes that are involved in the synthesis and modulation of the trypanothione redox system, including trypanothione reductase (TryR) and synthetase (TryS), and the enzyme dihydrofolate reductase (DHFR) that are thought to be essential, at least in *T. brucei* though, none of the identified inhibitors has been progressed to preclinical development. Further studies in this direction will hopefully lead to a breakthrough in terms of discovery of novel antitrypanosomal preclinical candidates. Among the many drug discovery strategies used to date, drug repurposing represents an expedited approach to develop innovative medications against neglected tropical diseases [7]. In this line, organizations such as the Medicines for Malaria Venture (MMV), through Open Source Drug Discovery programs, have made available repositionable compound libraries to facilitate and accelerate the search for new lead compounds against various diseases, including African trypanosomiasis [8]. Among the many available libraries, the Pathogen Box (MMVPB) library consists of 400 diverse drug-like molecules with known activity against tuberculosis, malaria, kinetoplastids, helminths, cryptosporidiosis, toxoplasmosis, and dengue [9]. Since the public launch of the MMVPB, it has been intensively investigated worldwide, leading to the discovery of active compounds against various pathogens including *Giardia lamblia* and *Cryptosporidium parvum* [10], helminths and barber’s pole worm [11], *Toxoplasma gondii* [12], *Trypanosoma brucei brucei (Tbb)*, *Trypanosoma cruzi*, *Leishmania donovani* [13] and *Echinococcus multilocularis* [14]. Similarly, we have explored the Pathogen Box with the aim of identifying promising starting points for drug discovery against trypanosomiasis. We herein present the activity profile of compounds emerging from the screening of the MMVPB against the bloodstream forms of *Trypanosoma brucei brucei*. A selected trypanocidal hit (MMV675968) reported to inhibit the dihydrofolate reductase (DHFR) enzyme was further explored for a preliminary structure–activity relationship study using analogs available at MMV. The most potent analogs were further analyzed in silico for potential inhibition of DHFR enzyme in trypanosomes, for time-kill kinetics, reversibility of trypanocidal effect, effect on parasite plasma membrane integrity, effect on reactive oxygen species, DNA fragmentation and ferric iron-reducing potency.

## 2. Results

### 2.1. Identification of Pathogen Box Compounds as Inhibitors of Bloodstream Forms of Trypanosoma brucei brucei

The primary screening of the 400 MMVPB compounds at a fixed concentration of 10 µM led to the identification of 70 (17.5%) compounds that inhibited the viability of trypanosomes by at least 90% (Figure 1). According to the MMVPB supporting information [9], seven of these hits were reference compounds, including two antimalarial drugs (mefloquine and primaquine) and five anti-trypanosomatid drugs, including two anti-HAT drugs (pentamidine and suramin), two anti-chagasic drugs (nifurtimox and benznidazole) and one antileishmanial drug (sitamiquine), thus validating the antitrypanosomal assay performed (Table 1). The remaining 63 active MMVPB compounds included, 25 inhibitors of the causative agents of kinetoplastids (*Trypanosoma* and *Leishmania* spp.) (Appendix A), 16 actives against tuberculosis, 14 against malaria, 3 against schistosomiasis, 2 against toxoplasmosis, 2 against cryptosporidiosis, and 1 against filariasis (Figure 1).

The 70 preselected compounds were submitted to a concentration–response study for the determination of their IC_50_ values. All compounds exhibited inhibition of *Trypanosoma brucei brucei* with IC_50_ values varying from 9.78 µM (MMV024311) to 0.0023 µM (MMV688180). All the 7 reference drugs were active (Table 1) with the anti-trypanosomal IC_50_ values ranging from 0.002 µM for MMV000062 (pentamidine- reference trypanosomiasis drug) to 8.4 µM for MMV000023 (primaquine-antimalarial drug). Among the 63 remaining compounds, 21 exhibited very high potency (IC_50_ ≤ 1 µM) among which compounds MMV688180 (IC_50_ = 2.3 nM) was by far the most active. Previous data indicate that this compound has high potency against kinetoplastids (*T. brucei brucei* and *T. brucei rhodesiense*) with IC_50_ < 0.13 µM (Pathogen Box supporting information). Additionally, Duffy et al., [13] reported the high potency of this compound (IC_50_ 0.01 µM) against *T. b. brucei*. A total of 31 compounds out of 63 displayed IC_50_ values between 1.0 and 4.0 µM, while 11 other compounds were moderately active (IC_50_ 4.38–9.78 µM) (Table 2 and Table 3). Overall, 25 actives (data not shown) out of the 63 hit compounds have known activity against kinetoplastids and thus were not further considered in this study. Thirty-eight (38) other compounds have known activity against different disease agents, but not kinetoplastids (Table 3 and Appendix A). Generally, the identified inhibitors displayed acceptable cytotoxicity profiles against the African green monkey kidney Vero cell line with selectivity indexes mainly greater than 10.

Among the 38 active compounds (Table 3) having known activity against different disease agents, not including kinetoplastids, compound MMV675968 (2,4 diaminoquinazoline) was the only inhibitor (IC_50_ 2.8 µM; SI > 35) with a small library of analogs available at MMV and was therefore selected for further investigation.

### 2.2. Preliminary Structure–Activity Relationship (SAR) Study with 23 Analogs of 2,4-Diaminoquinazoline (MMV675968)

Twenty-three (23) analogs of compound 2,4-diaminoquinazoline were available and donated by MMV for activity and selectivity testing against *T. b. brucei* and Vero cells. The results achieved, portraying the SAR of compounds are summarized in Table 4. Out of the 23 analogs, 11 compounds were inactive at concentrations ≤ 10 µM, and 12 exhibited activities with IC_50_ ≤ 4 µM of which five were more active than the parent compound (IC_50_ 0.56–0.045 µM). From the SAR analysis, analogs **2**–**5** bearing different substitution patterns around the phenyl moiety of compound **1** while keeping unchanged the 5-chloro-2,4-diaminoquinazoline core displayed an average IC_50_ value of 2.6 µM, similar to that of parent compound **1** (IC_50_ 2.6–2.8 µM). Concurrently, these substitutions on analogs **2**–**4** led to an increase in cytotoxicity by ~5-fold as compared to the parent hit with SI dropping from >37 down to 6.4. For analog **5**, the substitutions did not significantly impact on potency and selectivity (IC_50_ = 2.3 µM; SI > 43.47) compared to the parent hit compound (IC_50_ = 2.6 µM, SI > 37.87). Similarly, following the incorporation of an amide functionality in analogs **2**, **3**, **4**, **5**, the resulting analogs **11**–**15** exhibited IC_50_ values similar or higher than that of the parent compound (IC_50_ 2.5 to ˃10 µM), suggesting that neither the substitution pattern of the phenyl nor the amide group are essential for the antitrypanosomal activity. On the other hand, except for compound **11** (SI 5.60), hits **13**&**14** of this series exhibited selectivity indices greater than 30. Additionally, removal of the chlorine atom from the core moiety resulted in 4-fold (compound **6**—IC_50_ 0.56 µM) to 53-fold (compound **7**—IC_50_ 0.06 µM) increase in antitrypanosomal potency (series **6**–**9**) with respect to their corresponding congeners **3** and **2** (Series **1**–**5**) while concurrently improving their safety profile by 3–10 folds (SI 115–412) as compared to the parent compound **1**. In contrast, the introduction of chlorine atoms not on the 2,4-diaminoquinazoline core moiety but at positions 2 and 5 of the phenyl group afforded the most active and selective compound identified in this study, analog **10** (IC_50_ 0.045 µM; SI 1737). Unfortunately, the replacement of the 2,4-diaminoquinazoline core by the 2,4-diaminopyrimidine core moiety led to a total loss of activity (series **16**–**24**, IC_50_ ˃ 10 µM).

Overall, SAR studies on the parent hit **1** (MMV675968) (IC_50_ 2.6–2.8 µM; SI > 37) allowed the improvement in the in vitro potency and safety of analogs against bloodstream forms of *Trypanosoma brucei brucei* with up to ~40–58-fold increase in activity and selectivity with analogs **7** and **10** bearing the higher promise (IC_50_ 0.06 and 0.045 µM and SI 412 and 1737, respectively) (Table 4). In addition, analogs **7** and **10** duly fitted the lead-like properties, as stipulated in Lipinski’s rule of five (Figure 2).

### 2.3. In Silico Exploration of Trypanosomal DHFR and TR Inhibition by Analogs **7** and **10**

Literature data indicate that the diaminoquinazoline core is a suitable ligand for protein inhibition, including parasite enzymatic targets. More specifically, the quinazoline core was reported as a good motif for the inhibition of trypanothione reductase (TR) [16,17], a validated therapeutic target for antitrypanosomal drug development [18]. Moreover, several studies reported the efficacy of the 2,4-diminoquinazoline analogs against the enzyme DHFR, which is also a validated drug target in trypanosomes [19]. Specifically, the parent hit compound MMV675968 was previously reported to have antifolate activity. Indeed, Rosowsky et al. [20] explored the in vitro activity of 10 synthesized different analogs of 2,4-diamino-5-chloroquinazoline, including the hit compound MMV675968 (referred to as: 2,4-diamino-5-chloro-6-[(2,5-dimethoxyanilino)methyl]quinazoline) against DHFR enzymes of both *Pneumocystis carinii* and *Toxoplasma gondii* and determined IC_50_ values of 0.051 µM and 0.03 µM, respectively. In another more recent study, Nelson and Rosowsky identified compound MMV675968 (IV.18) as a highly potent inhibitor of the *Cryptosporidium parvum*-I DHFR in vitro with an IC_50_ of 0.0065 µM. This inhibition was further confirmed using phenotypic whole parasite testing with an IC_50_ of 0.2 µM [21]. Based upon this rationale, the two potent antitrypanosomal analogs of MMV675968 (MMV1578467 and MMV1578445) identified in this study were submitted to in silico screening against DHFR and TR as their potential targets, using molecular docking and molecular dynamics simulations.

#### 2.3.1. Accessible Protonation States of the Compounds and Molecular Docking

While molecules containing ionizable groups such as amines and carboxylates are stored in databases as neutral entities, they are mostly ionic under physiological conditions. For instance, amines become protonated to the quaternary form while carboxyl and other acidic groups such as phosphates and sulfates or hydroxylamines are deprotonated [22,23]. This has implications on in silico screening experiments as the protonation state tends to influence the binding strength and pose of a ligand within a binding pocket [22,24]. To account for the protonation states of compounds **7** and **10** at physiological pH, the dimorphite-DL [24] program and MolGpka [25] webserver were utilized as detailed in the Section 4. This revealed that positions 2 and 4 of the 2,4-diaminoquinazoline moiety of both compounds were ionizable at physiological pH [compound **7**: position 2 (pKa = 7.1) position 4 (pKa = 6.5); compound **10**: position 2 (pKa = 7.0), position 4 (pKa = 6.5)]. Hence, a total of three states were considered for each of the compounds–unprotonated (cmpd_**7**_unprot, cmpd_**10**_unprot), protonated at position 2 (cmpd_**7**_prot_2, cmpd_**10**_prot_2), and protonated at position 4 (cmpd_**7**_prot_4, cmpd_**10**_prot_4). Post docking analyses revealed that, in DHFR, all the accessible protonation states of both compounds shared similar poses within the same ligand recognition site in the binding pocket. This resulted in hydrogen bond formation between the diaminopyrimidine moiety and key DHFR ligand recognition residues [26,27,28], including the highly conserved Val32 (Figure 3A,B). Unlike DHFR, the TR enzyme consists of a wide binding site, where substrates and inhibitors have been shown to adopt different conformations with stacking observed for some inhibitor binding poses [29,30]. While both compounds **7** and **10** are bound to similar sites within the binding pocket of DHFR (Figure 3A,B), in TR, compound **7** is seen binding farther away from the cofactor FAD compared to compound **10** (Figure 3C,D). Furthermore, the docking scores revealed that both compounds had relatively stronger binding in DHFR compared to TR as follows–cmpd_**7**_unprot −8.8 vs. −8.2; cmpd_**7**_prot_2 −8.9 vs. −8.4; cmpd_**7**_prot_4 −9.2 vs. −8.2; cmpd_**10**_unprot −9.1 vs. −8.1; cmpd_**10**_prot_2 −9.1 vs. −8.3; and cmpd_**10**_prot_4 −9.2 vs. −8.0,—for DHFR and TR, respectively.

#### 2.3.2. Ligand Conformational Refinement and Binding Stability Monitoring through Molecular Dynamics Simulations

The dynamicity of drug binding and molecular recognition is only partially accounted for—through ligand flexibility—during docking [31]. Thus, MD simulation of the protein-ligand complex is required to further refine the predicted pose from docking and to ascertain the stability of binding. Here, 100 ns of all-atom MD simulations were performed on all the eight predicted poses from docking alongside the holoenzymes, making a total of eight systems (i.e., DHFR and TR bound to cmpd_**7**_prot_2, cmpd_**7**_prot_4, cmpd_**7**_unprot, cmpd_**10**_prot_2, cmpd_**10**_prot_4, cmpd_**10**_unprot, DHFR_holo and TR_holo). Plots of protein RMSD and Rg were used to check for convergence of the different trajectories between the ligand unbound and bound states. While the number of hydrogen bonds formed by a ligand within the binding site of an enzyme is crucial for its stability, monitoring the ligand RMSD across the simulation reveals the post docking dynamics of the bound ligand, which informs on the stability and strength of binding of the ligand [27]. To monitor the stability of binding of the different compounds, ligand RMSD with respect to protein structure and hydrogen bond numbers were computed across the different simulations. This revealed stable binding for all the DHFR-bound compounds, forming 1–7 hydrogen bonds (Figure 4). The total number of hydrogen bonds formed tended to increase with protonation, such that the unprotonated form had the lowest maximum number of hydrogen bonds compared to the protonated form of the compounds (Figure 4A). Unlike the DHFR-bound compounds, TR-bound compounds generally portrayed reduced stabilities. For instance, cmpd_**7**_prot_4 and cmpd_**10**_prot_2 lost hydrogen bonds completely during the simulation, with the later exiting from the binding site (Figure 4B). On the other hand, although cmpd_**10**_prot_4 maintained some hydrogen bonding during the simulation, its RMSD values varied considerably, pointing to unstable binding. It has been shown that microenvironmental differences within the binding site of an enzyme can influence its preference for binding to certain protonation states of a compound [32]. Thus, it is possible that while compounds **7** and **10** are good binders of DHFR in all their accessible protonation states, both compounds may have difficulties binding to and staying within the binding site of the TR enzyme, with only their unprotonated forms and the position 2 protonated compound **7**, having the possibility of binding successfully to the enzyme.

To further elucidate the key residues responsible for ligand recognition in both enzymes, ligand clustering of the last 10 ns of the MD trajectories was conducted as outlined in the Section 4. In DHFR, all the ligands populated a single cluster, further supporting the stability of binding of both compounds to DHFR in all their protonation states. This, however, was not the case with TR, where multiple clusters were populated particularly for the protonated forms of compound **10**. Examination of the ligand interaction patterns of the representative structures from the populated single clusters revealed the residues Val32, Asp54, Ile160 as key interacting residues, forming hydrogen bonds with the diaminoquinazoline (DMQ) moiety of both compounds in DHFR (Table 5). The rest of the residues mainly formed stacking and other interactions. In TR, however, no key residues were seen cutting across all the systems in terms of interaction. This may be expected due to the wide nature of the TR binding pocket as explained above.

### 2.4. Preliminary In Vitro Validation of the DHFR as MMV1578467 and MMV1578445 Target

#### 2.4.1. Protein Quantification and Confirmation

The total protein content of the prepared *T. b. b.* crude lysate was determined using the Bradford method. From a starting cell density of 5 × 10^9^ cell/mL, 7.7 mg/mL protein concentration was obtained. In order to confirm the protein extraction, SDS-PAGE profiling indicated successful extraction of proteins with molecular weights ranging from 3.5 to 240 kDa (Figure 5). The SDS-PAGE profile indicated a band resolving at an MW of approximately 25 kDa which likely corresponds to a complex containing the dihydrofolate reductase (DHFR) enzyme [33].

#### 2.4.2. Enzyme Activity and Inhibitory Effect of Compounds

Enzyme activity is usually estimated by measuring the rate of consumption of the substrate or the rate of product production over a given time and expressed as specific activity. In this study, the spectrophotometric evaluation of protein activity depended on the decrease in absorbance at 340 nm pertaining to the oxidation of NADPH cofactor in the presence of folic acid and DHFR. Hence, results from the estimation of the specific activity of the predicted DHFR contained in *T. b. b.* lysate are shown in Figure 6.

The results indicated a constant decrease in optical densities at 340 nm over 20 min, denoting the disappearance of NADPH in the reaction mixture, and thereby suggesting the presence of the predicted dihydrofolate reductase enzyme in the *T. b. b.* crude protein extract. This disappearance of NADPH was very likely correlated with enzymatic folate reduction. However, the achieved specific activity of 0.0964 U/mg (Figure 6A–E) was considerably lower compared to those reported for purified DHFR enzymes from recombinant *E. coli* [34] and *Leishmania major* [35]. Obviously, the portrayed activity by the crude protein extract is not attributable at 100% to the DHFR enzyme as some other enzyme (or multiple enzymes) that uses/use NADPH could be inhibited. Interestingly, the tested inhibitors displayed an inhibitory activity at 20 µM as depicted by the linear profile of their graphs similar to that of the background control that contained no protein extract (Figure 6A–D). Thus, as a consequence of inhibitors’ action, there was a significant (*p* < 0.05) drop in specific activity as compared to the enzyme in absence of inhibitors (Figure 6E). The investigated inhibitors were subsequently submitted to concentration–response studies to determine their median inhibitory concentration (IC_50_). As depicted in Figure 7, compound **1** (MMV675968), analog **7** (MMV1578467), analog **10** (MMV1578445) and methotrexate (reference DHFR inhibitor) inhibited *T. b. b.* crude protein extract in a dose-dependent manner with respective IC_50_ values of 0.4, 0.18, 2.45 and 0.1 µM.

### 2.5. Tentative Elucidation of Other Modes of Action of Analogs **7** and **10** against T. b. brucei

Based on their in vitro activity and selectivity profiles and their suitable drug-likeness, compounds **7** and **10** were chosen for further study of their tentative mode of action against *T. b. brucei*.

#### 2.5.1. In Vitro Kinetics of *T. b. brucei* Killing upon Treatment with Analogs **7** and **10**

Analogs **7** and **10** were assessed at their respective IC_99_, IC_90_, IC_50_ and IC_10_ for their impact on the growth rate of *Trypanosoma brucei brucei* in culture (Figure 8). The results indicated a concentration-dependent reduction in trypanosome growth (Figure 8A,B) compared to untreated cells when exposed to compounds **7** and **10** at their IC_99_, IC_90_, IC_50_ and IC_10_. A similar trend was observed for the positive control (pentamidine—Figure 8C). Of particular interest, compound **10** completely suppressed the growth of trypanosomes throughout the 72 h incubation period, whereas the effect of compound **7** tended to decrease after 60 h of treatment (Figure 8B). Inhibitor removal at 4 h for analog **10** and 24 for analog **7** when tested at their respective IC_90_ and IC_99_ followed by subculture in inhibitor-free complete medium indicated a cidal effect for compound **10** (Figure 8D). Conversely, at the time of compound **7** removal, the parasites started growing consistently between 24 and 48 h, followed by exponential growth between 48 and 60 h, denoting a rather static effect (Figure 8E). Of note, pentamidine (positive control) portrayed a static profile because its effect started diminishing increasingly from 48 h to 72 h (Figure 8F). Overall, from the comparison of the individual effects of the 2 inhibitors and pentamidine depicted in Figure 8G,H, analog **10** appeared to exhibit a more potent (cidal) effect on trypanosomes than analog **7** and pentamidine.

#### 2.5.2. Effect of Compounds **7** and **10** on Plasma Membrane Integrity

To evaluate their possible effect on plasma membrane integrity, various concentrations (IC_99_, IC_90_, IC_50_, and IC_10_) of compounds **7** and **10** were incubated with parasites for up to 120 min. Plasma membrane disruption was examined using the fluorescent probe SYBR Green, which binds to DNA and follows parasite injury permeability caused by direct or indirect action of the compounds. The results indicated that the parasite membrane conserved its integrity even after 120 min of contact with inhibitors. This unchanged status of the plasma membrane was materialized by a nonsignificant difference in the fluorescence profile of drug-treated versus untreated parasites. Conversely, there was a highly significant difference between the effects of compounds **7** and **10** on the one hand and the positive control (saponin) on the other hand, at *p* < 0.05 (Figure 9), denoting no impairment or a rather mild effect of the test compounds on the plasma membrane permeability of the parasite.

#### 2.5.3. Induction of Oxidative Stress in Trypanosomes by Compounds **7** and **10**

We assessed the influence of the test compounds on reactive oxygen species (ROS) production by trypanosomes as an indication of oxidative stress induction. Trypanosomes were incubated with compounds **7** and **10** for 120 min, and the intracellular level of ROS was determined using the fluorescent probe H_2_DCF-DA. The results obtained indicated no significant change (*p* < 0.05) in ROS production in treated parasites when compared to the untreated parasites (negative control), contrary to those treated with the positive control (H_2_O_2_) at 0.1% (*v*/*v*) (Figure 10).

#### 2.5.4. Compound **10** Induces DNA Fragmentation in Trypanosomes as a Sign of Cell Death

To shed light on the mode of parasite killing induced by the inhibitors, we further determined the type of cell death elicited by compounds **7** and **10** using a DNA fragmentation kit. The technique used consisted of immunological detection of BrdU (5′-bromo-2′-deoxyuridine)-labeled DNA fragments into the parasitic cytoplasm for apoptosis and into the culture supernatant for cell-mediated cytotoxicity. The results achieved are summarized in Figure 11 below.

Mediated cell death was depicted through the measurement of stronger absorbance emitted by DNA-labeled fragments with a characteristic green color. From the examination of Figure 11A, there was no apparent induction of cell lysis by any of the test compounds, similar to pentamidine relative to Triton-X, which is considered a potent cell disruptor. Concerning apoptotic DNA fragmentation, compound **10** exhibited a significant inducing effect. However, this effect was to a great extent lower than that exhibited by pentamidine, the reference trypanosomiasis drug (Figure 11B), at their respective IC_50s_. Of note, mediated apoptosis was not seen for all inhibitors at their IC_99_, IC_90_ and IC_10_, probably because the parasite death rate at IC_99_ and IC_90_ was significantly high to enable the development of the measurable green anti-BrdU-DNA complex, or the parasite killing rate was not significant at IC_10_ (due to subinhibitory drug concentration) to enable measurement of the drug effect. It should also be pointed out that compound **7** did not exert any effect through the two cell-mediated death modes investigated.

#### 2.5.5. Ferric Ion Reducing Antioxidant Power (FRAP) of Inhibitors

To assess the potential of inhibitors as reducing agents, we determined their ferric iron reducing capacity. The results showed that compound **7** exhibited weak Fe^3+^-reducing activity with a median reducing concentration (RC_50_) of 96.37 µM. In addition, compound **10** (RC_50_ ˃ 400 µM) was over ~4-fold less active than compound **7**. The latter showed more potent Fe^3+^-reducing activity than ascorbic acid (165 µM), which was used as a positive control. This finding suggests that compound **7** might exert its antitrypanosomal activity through the deprivation of ferric iron bioavailability to trypanosomes.

### 2.6. Prediction of ADMET Properties of Selected Inhibitors

It is known that a promising pharmacological activity of a particular compound does not guarantee its successful development because it should meet the suitable pharmacokinetic properties requirement. To determine whether the selected compounds are likely to be further developed as antitrypanosomal agents we have performed a computational analysis of their ADMET properties (absorption, distribution, metabolism, elimination, and toxicology) (Appendix A). Pharmacokinetic analysis revealed that all tested inhibitors meet the requirements of Lipinski’s, Ghose, Weber, Egan, and Muegge rules of drug-likeness. They were predicted to be well absorbed in the human intestine and to have a moderate water solubility. Unfortunately, the computational data do not indicate the target compounds as capable of crossing the BBB. The latter feature is a disadvantage if considering the possible CNS localization of parasites during the late-stage HAT. This therefore suggests that the compounds cannot be used for the treatment of the late meningo-encephalitic phase of the disease. All the tested compounds are substrates of P-gp, meaning that they might be expelled out of the cells following absorption. They all can be metabolized through the cytochromes 2D6, 2C9, 3A4 and 1A2. Regarding toxicity, none of the compound was hepatotoxic. However, they were found highly probable for immunotoxicity. Globally, these predictive evaluations give insights into the acceptable ADMET parameters of the tested compounds thus encouraging the lead optimization for a future drug development.

## 3. Discussion

Sleeping sickness remains a dreadful public health emergency, particularly in endemic regions where it causes significant damage to both cattle and humans [1]. Of course, control measures have been established for decades and include chemotherapy, which consists of the use of approved drugs. However, these drugs have several drawbacks, such as cumbersome lengthy treatment, adverse and toxic effects, and the development of resistant trypanosome mutants [4,6,36]. This situation emphasizes the need for more attention to this neglected tropical disease (NTD) in view of definitely eliminating it. Consequently, new therapeutic options are urgently needed to supply the pipeline of antitrypanosomal drug discovery and development, which might be suitable for optimal management of trypanosomiasis. To achieve this goal, many strategies have been developed to date and have gained successful results. This is the case for drug repositioning, which offers a tangible opportunity to quickly identify new chemical entities with appropriate potency against pharmacologically validated targets in trypanosomes [7].

Within this framework, we screened the Medicines for Malaria Venture’s Open Access Pathogen Box against the bloodstream forms trypomastigotes of *Trypanosoma brucei* subsp. *brucei*, Lister 427 VSG 221. Out of the 400 compounds tested, 70 were found to inhibit the growth of the parasite by at least 90% (Figure 1), and their IC_50_ values ranged from low micromolar (~9.8 µM) to low nanomolar (IC_50_~2 nM). Veale and Hoppe [37] obtained approximately similar results from the screening of MMVPB, with 65 hits identified against *T. b. brucei*, with >80% growth inhibition at 20 µM. Additionally, 21 of our identified hits did not match with those selected in their study. Similarly, while screening the MMVPB at 10 µM against the same parasite, Duffy et al., [13] reported 95 active compounds with a 50% inhibition cutoff. Cross examination of the data generated spotted 46 common hits identified, and 49 compounds different from those found in the present study. Additionally, 13 hits were selected in this study that were not reported by these authors. The observed discrepancies in activity profiles are probably due to differences in the culture and assay conditions used including, but not limited to incubation period (72 h here versus 48 h for Veale and Hoppe), parasite load (2 × 10^5^ parasites/mL here versus 2.4 × 10^4^ parasites/well-Veale and Hoppe [37]. From the 70 compounds that emerged from our preliminary screening, internal controls (reference compounds) were excluded from further studies, including five anti-trypanosomatids (MMV637953 (Suramine)—Trypanosomiasis; MMV001499 (Nifurtimox)—Chagas disease; MMV688773 (Benzimidazole)—Chagas disease; MMV000062 (Pentamidine)—Trypanosomiasis; MMV000063 (Sitamaquine)—Leishmaniasis) and two antimalarials (MMV000016-Mefloquine and MMV000023-Promaquine). Another set of 25 other compounds previously reported to inhibit trypanosomatid parasites was also identified (Table 2). Selected examples among these included compound MMV688180, a benzenesulfonamide (IC_50_ 0.0023 µM) that was the most active against *T. b. brucei* (Table 2). This compound was previously reported for its activity toward trypanosomes via the inhibition of N-myristoyl transferase, an enzyme that is essential for the survival and virulence of *T. b. brucei* [38,39]. Additionally, MMV689029 and MMV689028, which form part of the benzyl piperazine class of compounds, and the 2,4-substituted furan MMV688796 were previously reported by Duffy et al. [13] as promising starting points for drug discovery against both *T. cruzi* and *T. brucei*. Similarly, the kinase inhibitor MMV676604 (a 2-aminopyramidine), which exerts trypanocidal activity through inhibition of *Tb*ERK8 (extracellular signal-regulated kinase 8), was reported by Valenciano et al. [40]. MMV652003, which is a member of the benzamide class of molecules, has also been reported to act on leucyl-tRNA synthetase, a pharmacologically significant target of which is the trypanosome [41]. The butyl sulfanilamide MMV688467 was proven to inhibit microtubule formation in trypanosomes [42]. As predicted for other guanidine derivatives, we can hypothesize that the antiplasmodial MMV688271 [43] binds to the DNA minor groove at AT-rich regions [44]. All these compounds were equally excluded from further investigations.

The other set of 38 antitrypanosomal hits identified in this study were previously reported for activity against other disease targets, including malaria (14), tuberculosis (16), toxoplasmosis (2), schistosomiasis (3), cryptosporidiosis (2) and filariasis (1) (Table 3). Among these compounds, we opted to investigate the SAR of MMV675968 (2,4 diaminoquinazoline). This option was supported by the generous donation of a small library of 2,4-diaminoquinazoline analogs available at MMV. Of note, the data generated from our work indicated that compound MMV675968 has a favorable profile for further studies, including an antitrypanosomal IC_50_ of 2.8 µM and a selectivity index >35. Interestingly, this hit compound was previously reported to strongly inhibit the dihydrofolate reductase (DHFR) enzyme in *Cryptosporidium*, [21], *Pneumocystis carinii* and *Toxoplasma gondii* [20]. More recently, compound MMV675968 emerged as the most active pathogen box compound from a screening against *Toxoplasma gondii* (IC_50_ 0.02 µM; SI 275) [12]. Additionally, it has shown potency against other pathogens, such as planktonic forms of *C. albicans* [45] and *P. falciparum* (IC_50_ 0.07 µM) [13]. This strong rationale, added to the favorable pharmacological (IC_50_ 2.6/2.8 µM; SI > 35/37) and physicochemical (Figure 2) profiles of MMV675968, motivated the SAR study of its analogs. The antitrypanosomal pharmacomodulation study of MMV675968 (Table 4) revealed that activity varied consistently according to the various substituents added to the core 2,4-diaminoquinazoline structure and the adjacent phenyl portion. For instance, a change in the position of the methoxy substituents in analogs **2**, **3**, **4** and **5** did not significantly influence (*p* > 0.05) the activity compared to parent hit **1**. Chlorine removal from the core structure appeared to be beneficial for antitrypanosomal activity and selectivity, as the resultant nonchlorinated analogs **6**, **7**, **8**, and **9** were more potent and selective than their corresponding chlorinated congeners **2**, **3**, **4**, and **5**. Of note, this chlorine withdrawal on the core structure led to the identification of analog **7** (6-(((2-methoxyphenyl)amino)methyl)quinazoline-2,4-diamine), which was ~43-fold and ~11-fold more active and selective, respectively, than the parent hit **1**. Additionally, analog **7** was ~53-fold more active and ~64-fold more selective than the closest analog **3**. Conversely, chlorination of the phenyl portion of the parent hit (**1**) at positions 2 and 5 led to analog **10** (6-(((2,5-dichlorophenyl)amino)methyl)quinazoline-2,4-diamine) that exhibited outstanding ~58-fold and ~46-fold increases in activity (IC_50_ 45 nM) and selectivity (SI ~1737) *p* < 0.0001). Previously, Iwatsuki et al. [46] reported an improved antitrypanosomal activity by up to 54-fold for chlorinated antibiotic derivatives compared to nonchlorinated analogs. However, it is noteworthy that depending on the chlorination point on the core or adjacent portion, the chlorine atoms may confer antitrypanosomal activity to the afforded compound. In this line, compound **7** is the most attractive in terms of lipophilic efficiency. Indeed, swapping one methoxy for two chlorine substituents in compound **10** increase lipophilicity and might explain the marginal improvement in potency. On another note, polarizing the bond between the phenyl ring and the quinazoline core resulted in a drastic loss in activity (*p* < 0.0001), as evidenced by increased IC_50_ and mild selectivity values for analogs **11**–**15** (IC_50_ 2.5 to >10 µM) when compared to the profile of close analogs **2**, **3**, **4** and **5**. Finally, complete loss of activity (IC_50_ > 10 µM) was observed following shrinkage of the quinazoline core in favor of pyrimidine (Analogs **16**–**24**). The discrepancies observed in the pharmacological properties of the 23 analogs of MMV675968 indicate that any modification around the quinazoline core may redefine its binding properties to the parasitic target of interest. Analogs **7** and **10** emerged from the SAR study as the more promising candidates. They were therefore prioritized and progressed for additional analyses. In silico deciphering of the effect of analogs **7** and **10** indicated that both compounds are potent binders of the DHFR enzyme, binding in all their accessible protonation states, and engendering interactions with key DHFR ligand recognition residues such as Val32, Asp54, and Ile160. In an attempt to provide an extent of validation to the depicted in silico binding to trypanosome DHFR enzyme of the 2 promising structural analogs of MMV675968 (**7**-MMV1578467 and **10**-MMV1578445), we have assessed their preliminary in vitro activity against *T. b. brucei* crude protein extract that very likely contains the DHFR enzyme. Although lower activity was demonstrated against *T. b. brucei* crude protein extract, the results nevertheless portrayed a consistent inhibition of the oxidation of the NADPH cofactor by the parent hit (MMV675968) and the two promising analogs (MMV1578467 and MMV1578445) (Figure 7). The effective but low activity compared to previously reported data [28,47] could be tentative justified by the fact that a crude protein extract was used in the assays rather than the purified DHFR enzyme. Future in vitro/in vivo studies using recombinantly purified DHFR enzyme and appropriate animal models of trypanosomiasis will enable us to confirm our assertions. Further studies on analogs **7** and **10** included deciphering their time-kill kinetics, induction of intracellular ROS production, membrane permeabilization, DNA fragmentation and ferric ion reducing antioxidant power (FRAP). Compound **7** was found to alter the growth of parasites after a period of 24 h at IC_99_ and presented a parasitostatic effect on trypanosomes, which was further confirmed by the reversibility of the drug effect after drug removal and subculture. We can therefore argue that this compound is either a slow-acting inhibitor or might be involved in a reversible interaction with the trypanosome metabolic target of action. Such a drug might rapidly induce drug resistance selection and will require either multiple and high doses to cure the mice in an in vivo study or a very long treatment period. This profile of compound **7** is of limited advantage compared to the current remedies used for the treatment of trypanosomiasis. Of particular interest, compound **10** presented a fast-killing effect within 4 h and irreversible cidal activity during the whole monitoring period of 72 h at its IC_99_, contrary to compound **7**, and pentamidine (reference anti-trypanosomiasis drug) exhibited a gradually diminishing effect from 24–72 h post drug removal. This time-kill kinetics profile of compound **10** validates it as a promising candidate for further development against trypanosomiasis. Further attempts to understand the mode of action of compounds **7** and **10** against trypanosomes demonstrated that none of the compounds elicited deterrent effects on the plasma membrane permeability of the parasite. Additionally, no induction of a significant imbalance in intracellular ROS levels was observed compared to untreated parasites. Therefore, we suggest that membrane permeability and oxidative stress do not contribute to the mechanism of action of compounds **7** and **10** against *Trypanosoma brucei brucei* parasites. Moreover, exploration of the mode of elicited cell death indicated that the compounds did not induce cell death through cytolysis. This finding corroborates the absence of a deterrent effect by compounds **7** and **10**, as previously demonstrated in our membrane permeability assay. Subsequent detection of DNA fragments in the cell lysate of parasites exposed to compound **10** confirmed its apoptosis-like-inducing effect through elicitation of DNA fragmentation in treated bloodstream trypanosomes materialized by a significant increase in absorbance compared to the negative control. Similarly, the positive control (pentamidine) displayed a high DNA fragment signal. Of note, pentamidine has been previously reported to have specific and strong DNA-binding properties, particularly to the minor groove of AT-rich regions [48]. This finding further validates the approach using the DNA fragmentation ELISA kit in our study. Moreover, many other reports have previously mentioned the apoptosis induction potency of pentamidine against Kinetoplastidae [49,50]. Further investigations are warranted to obtain more insights into the mechanism of action by which compound **10** induces cidal effect in trypanosomes. More specifically, it is important to determine whether the inhibitor acts on the kinetoplast or on nuclear trypanosomal DNA and to determine the different biochemical and morphological changes that occur in parasites treated with compound **10**. Finally, the assessment of the ferric iron-reducing ability of the inhibitors showed moderate reducing power by compound **7**, while compound **10** showed no activity. Iron is a vital element in most living organisms, including trypanosomes, and is involved in several important biological processes, such as mitochondrial respiration, DNA replication, antioxidant defense, and glycolysis. In fact, three enzymes were described as being iron-dependent and indispensable for trypanosomes. This is the case for superoxide dismutase, which eliminates superoxide radicals released during generation of the tyrosyl radical in the R2 subunit of ribonucleotide reductase [51,52]. Alternative oxidase is an important enzyme for the reoxidation of nicotinamide adenine dinucleotide (NADH) produced during glycolysis [53,54]. In addition, Ayayi et al. [55] investigated the iron dependence of oxidase alternative and terminal trypanosomes (AOT) by chelating iron using o-phenanthroline, which resulted in strong inhibition of this enzyme. Ribonucleotide reductase is another iron-dependent enzyme that catalyzes the reduction of ribonucleotides to deoxyribonucleotides needed for DNA synthesis [56,57]. Therefore, iron deprivation of parasites by compound **7** might induce a loss of viability of these vital enzymes, thereby resulting in a rapid decrease in DNA synthesis, increased oxidative stress and cessation of electron transfer to the AOT enzyme, thus contributing to the death of the parasite.

## 4. Materials and Methods

### 4.1. Compounds Handling and Storage

The MMVPB manufactured by Evotec (USA) was obtained free of charge from the MMV (Geneva, Switzerland). The box consisted of 400 drug-like compounds, shipped on dry ice and supplied as five 96-well microtiter plates containing 10 µL of 10 mM stock solutions of compounds in 100% dimethyl sulfoxide (DMSO). Supporting information for compounds was found at https://www.mmv.org/mmv-open/pathogen-box (accessed on 4 July 2019) and included plate layout, chemical structures and formula, molecular weights, in vitro and in vivo DMPK, confirmed biological activities against some neglected disease pathogens and cytotoxicity data. Compounds were diluted to five subsets for a final intermediary concentration of 100 µM in 96-well storage plates using incomplete IMDM (Iscove’s modified Dulbecco’s medium) culture medium (2 µL of stock solution added to 198 µL of sterile incomplete medium). Plates were stored at −20 °C until biological assays. The diaminoquinazoline solid analogs were also provided by the MMV organization (Geneva, Switzerland) and were dissolved in DMSO to reach a concentration of 10 mM. Pentamidine isethionate (Sigma Aldrich) and podophyllotoxin (Sigma Aldrich) were weighed and dissolved in 100% DMSO to a final concentration of 10 mM and further diluted and used as positive controls for the antitrypanosomal and cytotoxicity assays, respectively.

### 4.2. Antitrypanosomal Screening of the Open Access MMV Pathogen Box

#### 4.2.1. Parasite Growth Conditions

The parasite used for this study was the bloodstream form trypomastigotes of *Trypanosoma brucei* subsp. *brucei*, Strain Lister 427 VSG 221 kindly donated by BEI resources (https://www.beiresources.org/ accessed on 4 April 2019). Parasites were axenically cultivated in sterile vented flasks containing complete Hirumi’s modified Iscove’s medium 9 (HMI-9) [500 mL IMDM (Iscove’s modified Dulbecco’s medium) (Gibco, Waltham, MA, USA) supplemented with 10% (*v*/*v*) heat-inactivated fetal bovine serum (HIFBS) (Sigma Aldrich), 10% (*v*/*v*) serum plus (Sigma Aldrich), HMI-9 supplement (1 mM hypoxanthine, 0.16 mM thymidine, 50 µM bathocuproine disulfonic acid, 1.5 mM cysteine, 1.25 Mm pyruvic acid, 0.2 mM 2-mercaptoethanol (Sigma Aldrich)), and 1% (*v*/*v*) penicillin–streptomycin (Sigma Aldrich) and incubated at 37 °C in a 5% CO_2_ atmosphere. Cultures were routinely monitored every 72 h using a Lumascope LS520 inverted fluorescence microscope (Etaluma, Inc., USA) to assess parasite density and subsequently passaged with fresh complete medium in such a way that the cell density never exceeded 2 × 10^6^ cells.mL^−1^ [58].

#### 4.2.2. In Vitro Single Point and Concentration–Response Antitrypanosomal Screening

The in vitro inhibitory potency of the 400 MMVPB compounds against bloodstream forms of *Trypanosoma brucei brucei* was evaluated using the resazurin-based inhibition assay as previously described [59]. Briefly, parasites at their mid-logarithmic growth phase were counted, and the cell density was adjusted with fresh complete HMI-9 medium to 2 × 10^5^ trypanosomes per mL. Then, 90 µL of parasite suspension was then distributed into the wells of 96-well flat-bottomed plates containing 10 µL of compounds for a final test concentration of 10 µM. The first and last columns in each plate served as negative (cells with 0.1% DMSO) and positive (cells with 10 µM pentamidine isethionate) controls, respectively. After 68 h of incubation at 37 °C and 5% CO_2_, parasite viability was checked after fluorescence measurement using a Tecan Infinite M200 fluorescence multiwell plate reader (Austria) at wavelengths of 530 nm for excitation and 590 nm for emission following a 4 h incubation period with resazurin (0.15 mg/mL in DPBS, Sigma–Aldrich) in darkness. Each assay plate was set up in duplicate and repeated two times. The percent parasite inhibition was determined for each compound based on fluorescence readouts relative to the mean fluorescence of negative control wells. Compounds exerting a mean inhibition percentage greater than 90% at 10 µM were selected and tested in duplicate at 5-point concentrations using the aforementioned conditions. Likewise, analogs of a selected hit were also tested. Mean fluorescence counts were normalized to percent control activity using Microsoft Excel, and the 50% inhibitory concentrations (IC_50_) were calculated using Prism 8.0 software (GraphPad) with data fitted by nonlinear regression to the variable slope sigmoidal concentration–response formula:

y = 100/[1+ 10^(logIC50/99-*x*)*H*^], where *H* is the hill coefficient or slope factor [60]. Prioritized compounds (IC_50_ < 4 µM) were further tested for their cell cytotoxic effect as described below.

### 4.3. Determination of the Cytotoxicity of Inhibitors against Vero Cells

#### 4.3.1. Maintenance of Mammalian Cells

The African green monkey kidney Vero cell line (ATCC CRL-1586) was grown in T-25 vented cap culture flasks using complete Dulbecco’s modified Eagle’s medium (DMEM) supplemented with 10% FBS, 1% nonessential amino acids and 1% (*v*/*v*) penicillin–streptomycin and incubated at 37 °C in an atmosphere containing 5% CO_2_. The medium was renewed every 72 h, and cell growth was assessed using an inverted microscope (Lumascope LS520). Subculture was performed when the cells reached ~80–90% confluence by detaching with 0.25% trypsin-EDTA followed by centrifugation at 1800 rpm for 5 min. The resulting pellet was resuspended and counted in a Neubauer chamber in the presence of trypan blue to exclude nonviable cells colored in blue. Once the cell load was estimated, they were either used for the next passage in a new flask or processed for the cytotoxicity assay.

#### 4.3.2. Assessment of the Cytotoxic Effect of Compounds

The cytotoxicity of promising compounds was assessed as previously described by Bowling et al. [59] in a 96-well tissue culture-treated plate. Briefly, Vero cells at a density of 10^4^ cells per well were plated in 100 µL of complete DMEM and incubated overnight to allow cell attachment. Plates were then controlled under an inverted fluorescence microscope (Lumascope LS520) to ensure adherence, sterility and cell integrity. Thereafter, culture medium from each well was carefully emptied, and plates were filled with 90 µL of fresh complete medium followed by the addition of 10 µL of serial 5-fold dilutions of compound solutions. Podophyllotoxin (100 µM–0.16 µM) and 0.5% DMSO (100% cell viability) were also included in assay plates as positive and negative controls, respectively. After an incubation period of 48 h at 37 °C in a humidified atmosphere and 5% CO_2_, 10 µL of a stock solution of resazurin (0.15 mg/mL in DPBS) was added to each well and incubated for an additional 4 h. Fluorescence was then read using a Magelan Infinite M200 fluorescence multiwell plate reader (Tecan) with excitation and emission wavelengths of 530 and 590 nm, respectively. The percentage of cell viability was calculated from readouts, and the median cytotoxic concentration (CC_50_) for each compound was deduced from concentration–response curves using GraphPad Prism 8.0 software as described above. Selectivity indexes were then determined for each test substance as follows: SI = CC_50(Vero cells)_/IC_50(*T. brucei brucei*)._

### 4.4. In Silico Exploration of the DHFR and TR Binding Properties of Analogs **7** and **10**

#### 4.4.1. Compound Preparation

Compound structures were drawn and converted to SMILES format using the smiles generator window of the cheminfo webserver (http://www.cheminfo.org/ accessed on 3 May 2022). Accessible protonation states of the compounds at physiological pH were assessed using the dimorphite-DL program [24]. Dimorphite-DL uses a straightforward empirical algorithm that leverages substructure searching and makes use of a database of experimentally characterized ionizable molecules to enumerate small-molecule ionization states [24]. Here, the SMILES format of the compounds was used as input, generating a list of SMILES of all the possible protonation states at the default physiological pH (6.4–8.4). The output from dimorphite-DL was streamlined by measuring the pKa values using MolGpka [25]. The MolGpka webserver predicts pKa through a graph-convolutional neural network model that works by learning pKa related chemical patterns automatically and building reliable predictors with the learned features [25]. Matching the pKa values of the different ionizable sites with the physiological pH resulted in the retention of only the physiologically feasible compounds. The retained SMILES were then converted to the vina compatible pdbqt format using the format converter window of the cheminfo webserver in preparation for docking.

#### 4.4.2. Protein Structure Preparation

Protein structures used in this study were obtained from the Research Collaboratory for Structural Bioinformatics, Protein Data Bank (RCSB PDB) [61]. For *Trypanosoma brucei brucei* trypanothione reductase, the crystal structure (PDB ID: 2WP6) is available and was downloaded. On the other hand, *T. brucei brucei* DHFR crystal structure has not been deposited in the RCSB PDB. However, it shares 100% sequence identity with *T. brucei rhodesiense* DHFR, hence the latter’s crystal structure (PDB ID: 3RG9) was downloaded and prepared for use. The protein structures were pre-processed using Discovery studio visualizer version 4.1 [62]. Initially, the structure of DHFR was examined for the presence of the recently identified DHFR crystal structural error, reported in *P. falciparum* [63]. The implicated loops were identified to be shorter and capable of no entanglements, hence ruling out the possibility of a crystallographic error. The cofactors NADPH and FAD were maintained in both the DHFR and TR structures, respectively. Partial charges and AutoDock atom-types (pdbqt format) were incorporated in the protein and cofactor structures, respectively using the *prepare_receptor4.py* and the *prepare_ligand4.py* python scripts from AutoDock4 tools [64].

#### 4.4.3. Molecular Docking

A blind docking protocol was implemented using the molecular docking and virtual screening tool AutoDock Vina [65]. Docking validation was accomplished through a redocking of the co-crystalized ligands of the DHFR and TR structures into their respective active sites. Docking parameters adopted following the validation were as follows: DHFR—box size (in Å) x = 42.75, y = 42.75, z = 41.62; box center x = 64.64, y = 32.86, z = 36.54; and an exhaustiveness of 290. TR—box size (in Å) x = 83.62, y = 65.62, z = 84.00; center x = 43.17, y = 5.75, z = −0.05; and an exhaustiveness of 290. After the docking a split of the top nine predicted poses from AutoDock vina was performed using the *vina_split* script and the pose with the lowest docking score was retained for further evaluations. Protein—ligand complexes were then prepared for the top scoring ligands using an in-house python script and visualization was carried out in Discovery studio visualizer version 4.1 and PyMOL [66].

#### 4.4.4. Molecular Dynamics Simulations

Molecular dynamics (MD) simulations were performed within the Amber forcefield a99SB-disp [67], using the GROMACS v.2018 software package [68]. The GROMACS compatible version of the a99SB-disp forcefield was obtained from [69] and used to perform all-atom MD simulations. Ligand parameters were determined by the ACPYPE tool [70]. TIP4P (a99SBdisp_water) water molecules were used to embed each system within a cubic simulation box, leaving a clearance space of 1.0 Å from the edges of the protein. Appropriate amounts of Na^+^ and Cl^−^ ions were added to neutralize the total system charges. This was followed by system relaxation through energy minimization using the steepest descent algorithm with a force threshold of 1000 kJ/mol/nm and a maximum of 50,000 steps. The temperature and pressure were, respectively equilibrated using the modified Berendsen thermostat (at 300 K for 100 ps), according to the NVT ensemble and the Parrinello–Rahman barostat [71], according to the NPT ensemble, to maintain the pressure at 1 bar. During the equilibration steps, the protein was position restrained and constraints were applied to all the bonds using the LINCS algorithm [72]. Finally, unrestrained production runs were performed for 100 ns each under periodic boundary conditions (PBC) and the equilibration step thermostat and barostat were both maintained for temperature and pressure couplings. The leap-frog integrator was used with an integrator time step of 2 fs, while the Verlet cut-off scheme was implemented using default settings, and coordinates were written at 10.0 ps intervals. Long-range electrostatic interactions were treated using the Particle-mesh Ewald (PME) algorithm [73], while short-range non-bonded contacts (Coulomb and van der Waals interactions) were defined at a 1.4 nm cut-off. All the analyses were accomplished using GROMACS tools. The trajectories were first corrected for periodic boundary conditions using the *gmx trjconv* tool—starting with system centring within the simulation box, fitting the structures to the reference frame, and putting back atoms within the box. Furthermore, *gmx rms*, *gmx rmsf*, and *gmx gyrate* were utilized for the calculation of the root mean square deviation (RMSD), root mean square fluctuation (RMSF), and radius of gyration (Rg), respectively. The *gmx_cluster* tool was used for ligand clustering, and the gromos method was used with an rmsd cut-off value of 0.12 nm.

### 4.5. In Vitro Screening of MMV1578467 and MMV1578445 against T. b. b. Crude Protein Extract

#### 4.5.1. Preparation of Crude *Trypanosoma brucei brucei* Lysates and Protein Quantification

Logarithmic phase *T. brucei brucei* bloodstream parasites were collected by centrifugation at 3500× *g* rpm, for 30 min at 4 °C (Allegra X-15R, Beckman Coulter, Brea, CA, USA), and washed twice in sterile DPBS 1X pH 7.4. The resulting cell pellets (from 5 × 10^9^ cell/mL inoculum) were lysed by adding 300 µL of RIPA lysis buffer consisting of 50 mM TrisHcl, pH 7.5, 150 mM NaCl, 1% Triton X-100 *v*/*v*, and 1% protease inhibitor cocktail. The homogenate was stirred for 30 min at 25 °C followed by 10 freeze (−80 °C)/thaw (42 °C) cycles to maximize lysis and extraction. Proteins extract was centrifuged at 14,000× *g* rpm (Eppendorf 5418-RG, Eppendorf AG, Hambourg, Germany) for 15 min at 4 °C, and the supernatant was collected and conserved at −80 °C until further use.

The protein concentration of the crude extract was determined by the Bradford method (Bradford protein assay kit 23200) using a bovine serum albumin standard curve [74]. Protein extraction was confirmed using sodium dodecyl sulfate-polyacrylamide gel electrophoresis (SDS-PAGE) (Owl Dual-Gel Vertical Electrophoresis systems, Thermo Scientific, Waltham, MA, USA) [75]. For this purpose, acrylamide separating and stacking gels were run at approximately 100 to 120 V during 1 h. At the end of the electrophoresis, gels were stained with Coomassie brilliant blue G-250 and destained with distilled water for visualization.

#### 4.5.2. *T. b. b.* Protein Activity Assay and Primary Screening of Inhibitors

In an attempt to estimate the folate reduction (dihydrofolate reductase activity), we monitored the overall decrease in absorbance at 340 nm due to the oxidation of NADPH and the reduction in folic acid into tetrahydrofolate (Figure 1) as previously described by Bailey and Ayling [76] with slight modifications.

Briefly, in a 96-well quartz microplate, 0.3 mg/mL of total protein from *Trypanosoma brucei brucei* lysate was preincubated for 1 min at 25 °C with 120 µM NADPH in an assay buffer containing 50 mM Tris Hcl, pH 7.5, 150 mM NaCl, 2 mM DTT. The assay was initiated by the addition of 50 µM folic acid (Sigma-Aldrich, Darmstadt, Germany) to yield a total reaction volume of 200 µL and the absorbance was measured at 340 nm (A340) in kinetic mode every 15 s for 20 min at room temperature. Methotrexate (Sigma-Aldrich, Darmstadt, Germany) which binds to and inhibits the enzyme dihydrofolate reductase was used as a positive control at 20 µM and was introduced before the addition of NADPH. Similarly, predicted DHFR inhibitors (compounds **1**, **7** and **10**) were screened at 20 µM in such a way that the DMSO concentration do not exceed 0.5%. Inhibitor control wells containing test compounds in the reaction buffer (as inhibitor background control to test the absorbance of the compounds alone) and solvent control wells containing protein extract and DMSO (0.2%) (to test the effect of DMSO on enzyme activity) were included in the assay. The background control wells consisting of NADPH and folic acid prepared in the same reaction buffer were also included. Enzymatic folate reduction activity was then measured as changes in A340 per min using the combined molar extinction coefficient for NADPH oxidation and folic acid reduction in Ɛ_[NADPH/THF]_ = 12.3 mM^−1^ cm^−1^. The enzyme-specific activity in the presence and absence of inhibitors was calculated using the formula:(1)$$\mathrm{Specific}\;\mathrm{activity}\;\left(\mathrm{Units}/\mathrm{mg}\;P\right)=\frac{\left[\left(\Delta \mathrm{OD}/\min_{\mathrm{sample}}-\Delta \mathrm{OD}/\min_{\mathrm{blank}}\right)\times \;\mathrm{df}\right]}{12300\;x\;V\;\times \;\mathrm{mg}\;\mathrm{protein}/\mathrm{mL}}$$
where ∆OD/min_blank_ represents the activity rate for the blank, ∆OD/min_sample_ is the activity rate for the enzymatic reaction, V is the protein volume in mL (the volume of protein extract used in the assay), df is the dilution factor of the protein extract, mg protein/mL is the protein concentration of the original sample before dilution, and Units/mg P is the specific activity expressed in µmole/min/mg protein.

#### 4.5.3. Determination of the IC_50_ Values of Test Compounds

The IC_50_ values of each test compound were determined by measuring the reaction rate (∆OD/min) at several inhibitor concentrations (20–0.00128 µM) as described above. The test was performed in duplicate and the results were normalized to those of the *T. b. b.* protein control containing no inhibitor (100% activity). The relative activity percentages were then deduced for each of the tested concentrations using the formula below.
(2)$$\%\;\mathrm{Relative}\;\mathrm{activity}=\frac{\mathrm{Specific}\;\mathrm{activity}\;\mathrm{of}\;T.b.b.\mathrm{Protein}\;\mathrm{with}\;\mathrm{inhibitors}\;}{\mathrm{Specific}\;\mathrm{activity}\;\mathrm{of}\;T.b.b.\mathrm{Protein}\;\mathrm{without}\;\mathrm{inhibitors}\;}$$

The 50% inhibitory concentrations (IC_50_) were deduced from a concentration–response curve using GraphPad Prism 8.0 software with data fitted by nonlinear regression to the variable sigmoidal concentration–response slope formula y = 100/[1 + 10^(logIC50/99-x)H^], where H is the hill coefficient or slope factor.

### 4.6. Attempts to Elucidate other Antitrypanosomal Modes of Action of Analogs **7** and **10**

#### 4.6.1. In Vitro Determination of Parasite-Killing Kinetics of Selected Hit Compounds

The preliminary structure–activity relationship study of compound MMV675968 (**1**) led to the identification of two highly potent and selective hits (analogs **7** and **10**). Thus, they were selected, and their effect at various concentrations was assessed on the proliferation rate of the bloodstream form of *Trypanosoma brucei brucei* using microscopic cell count. Briefly, parasites at their exponential growth phase were seeded into a 24-well flat-bottomed plate and incubated for 72 h at a cell density of 2 × 10^5^ trypanosomes per mL with compounds at their IC_99_, IC_90_, IC_50_, and IC_10_ values. After specified exposure time intervals (0, 4, 8, 12, 24, 30, 36, 48, 60, 72 h), the content of each well was harvested and counted on a Neubauer hemacytometer to determine the number of mobile parasites [77]. The obtained values were used to plot the growth curves as parasite density (cells/mL) versus incubation time using GraphPad Prism 8.0 software. All experiments were performed in duplicate and included positive (pentamidine) and negative (parasites without inhibitor) controls.

#### 4.6.2. Assessment of the Cidal or Static Effect of the Inhibitors

Data generated from the time-kill kinetics indicated that analogs **10** and **7** at their respective IC_99_ totally inhibited the growth of trypanosomes within 4 and 24 h, respectively. Therefore, the ability of parasites to recover post exposure to inhibitors **10** and **7** was further evaluated. Briefly, parasites at 2 × 10^5^ trypanosomes/mL were incubated at 37 °C and 5% CO_2_ with compounds at their respective IC_99_ and IC_90_ concentrations for 4 h and 24 h for compounds **10** and **7**, respectively. After that, the cells were washed three times with fresh complete medium by centrifugation at 2500 rpm for 7 min. Cell pellets were thereafter resuspended in fresh complete culture medium in a 24-well plate and subcultured under the same conditions for 68 h for compound **10** and 48 h for compound **7** to achieve the complete parasite growth pattern. Thereafter, cells were enumerated using the Neubauer cell counter at each time point. The assay was performed in duplicate, and the mean cell counts were plotted membrane against time using GraphPad Prism 8.0 to assess the cidal or static effect of inhibitors.

#### 4.6.3. Effect of Compounds **7** and **10** on Trypanosome Plasma Membrane Integrity

The plasma membrane is the first barrier protecting the cell from the external environment and therefore might represent an important challenge for compounds to exert their inhibitory action. Thus, compounds **7** and **10** were assessed for their ability to induce alterations in cell integrity as previously described [77] with a modification consisting of using an SYBR green assay [78]. The principle of this assay is based on the fact that upon treatment with membrane disruptors, trypanosomes with compromised plasma become fluorescent as a result of SYBR green entry and fixation to the exposed DNA. Practically, in a 96-well microtiter plate, 90 µL of trypanosome suspension (2 × 10^6^ cells/mL/well) was preincubated with SYBR Green (2X) (Sigma Aldrich) for 15 min at 37 °C and 5% CO_2_ in darkness. The reaction was then allowed to start following the addition of 10 µL of compounds at their IC_99_, IC_90_, IC_50_ and IC_10_. The plate was then incubated at 37 °C on a Magelan Infinite M200 multiwell plate reader (Tecan), and fluorescence was further recorded every 5 min for up to 120 min at λex = 485 and λem = 538 nm. Wells containing saponin (Sigma Aldrich) at 0.075 g/mL as a positive control for maximal permeabilization, untreated trypanosomes and HMI-9 medium representing the negative control and the background signal, respectively, were also included. The results are expressed as the mean ± SD of two experiments carried out in duplicate after deducting the background signal (wells containing HMI-9 medium) and then used to plot the fluorescence counts versus time (min) graph.

#### 4.6.4. Induction of Intracellular Reactive Oxygen Species (ROS) Production by Trypanosomes upon Treatment with Inhibitors

The production of ROS was detected using a 2′,7′–dichlorofluorescein diacetate (DCFDA) probe (Sigma–Aldrich) as described by Rea et al. [79]. Briefly, parasites (2 × 10^6^ cells/mL/well) were washed in incomplete IMDM medium and incubated with compounds **7** and **10** at their respective IC_99_, IC_90_, IC_50_ and IC_10_ for 2 h at 37 °C. DCFDA (100 µL, 5 µM) was then added to the parasites, and incubation was pursued for 15 min at 37 °C. Hydrogen peroxide (H_2_O_2_ 0.1% *v*/*v*) was used as a positive control for maximal ROS production, and wells with untreated parasites were included as a negative control. Of note, DCFDA is cleaved by ROS to produce fluorescent 2′,7′-dichlorofluorescein (DCF), of which the fluorescence intensity was measured at λex = 485 nm and λem = 520 nm. Data obtained from duplicate readouts were used to determine the ROS production percentage relative to the positive control (100% Production).
% ROS production = (OD Test/OD control(H2O2)) × 100

#### 4.6.5. Induction of Cellular DNA Fragmentation upon Trypanosome Treatment with Inhibitors

In an attempt to understand the tentative mechanism (apoptosis or cytolysis) of trypanosome death following exposure to compounds **7** and **10**, DNA fragmentation analysis was performed using a cellular DNA fragmentation ELISA kit according to the manufacturer’s recommendations (ROCHE Cat. No. 11585045 001 Sigma Aldrich). Basically, this assay tends to quantify apoptotic cell death by detection of fluorescent dye-labeled DNA fragments in the cytoplasm of affected cells or to measure cell-mediated cytotoxicity by detection of dye-labeled DNA fragments released from damaged cells into the culture supernatant. Briefly, parasites (4 × 10^5^ parasites/mL/well) were incubated at 37 °C for 24 h with BrdU (5′-bromo-2′-deoxyuridine), a nonradioactive thymidine analog that is incorporated into genomic DNA. BrdU-labeled parasites were harvested (by spinning at 2500 rpm for 7 min) followed by centrifugal washing using BrdU-free HMI-9 medium. One hundred microliters of the labeled and washed parasites at a final density of 4 × 10^5^ parasites/mL/well were treated in duplicate wells of a 96-well round-bottom plate with 100 µL of analogs **7** and **10** at their respective IC_99_, IC_90_, IC_50_ and IC_10_ for 4 h. The plate was then centrifuged at 2500× *g* rpm for 7 min, and the pellets were collected and kept at 4 °C until further use. For cytolysis measurement, the supernatants were added to a microplate coated with anti-DNA antibody to allow DNA capture from the test samples. The captured DNA fragments were subsequently denatured using microwave irradiation (LG NeoChef Charcoal Healthy Ovens) at 500 W for 5 min to separate DNA strands in view of displaying the BrdU. Thereafter, an anti-BrdU-antibody-POD (peroxidase) conjugate was added to detect the BrdU contained in the captured DNA fragments. One hundred microliters of the POD substrate solution were added to each well, and the absorbance was read at 370 nm every 30 s until color development, which was green in the case of this study.

For apoptosis measurement, the cell pellets from each well were resuspended in 200 µL of the kit’s incubation buffer and incubated for 30 min. The lysed cells were centrifuged at 1700× *g* rpm for 10 min, and the supernatants were used as described above. Triton X-100 (Sigma Aldrich) was used as a positive control for both apoptosis and cell-mediated cytotoxicity. The absorbance values were proportionally correlated to the amount of DNA fragments in the treated cultures. Data are expressed as the mean ± SD of two independent experiments performed in duplicate and compared to the negative control (untreated parasites) at a significance level of *p* < 0.05.

#### 4.6.6. Determination of the Ferric Ion Reducing Antioxidant Power (FRAP) of Inhibitors

Iron is one of the vital elements for all parasites, including trypanosomes, as it plays a key role in pathogenesis and the host immune response. Therefore, iron bioavailability reducing agents to the parasite could be potential candidates for drug development against trypanosomiasis. To this end, we determined the capacity of the test compounds to reduce iron from ferric to ferrous status using the method described by Benzie et al. [80]. Briefly, 25 µL of each compound (**7** and **10**) was added to 25 µL of a solution of Fe^3+^ prepared at 1.2 mg/mL in distilled water in a 96-well microplate. The plate was incubated for 15 min at room temperature in darkness, after which 50 µL of ortho-phenanthroline (0.2% in methanol) was added to achieve final compound concentrations ranging from 400 µM to 0.19 µM. Plates were reincubated for 15 min, and the absorbance was measured at 510 nm. Ascorbic acid was used as a positive control and tested at concentrations ranging from 100 to 0.048 μg/mL. The median reducing concentration (RC_50_) values of compounds were determined through sigmoidal concentration–response curves using GraphPad Prism version 8.0 software.

### 4.7. Prediction of ADMET Properties of the Hits

The structures of the test compounds (MMV675968, MMV1578467, MMV1578445) were designed using ChemBio2D Draw and their SMILES codes were generated. These codes were used as the main materials for running the online SwissADME (http://www.swissadme.ch/ accessed on 12 September 2022) and Protox (https://tox-new.charite.de/protox_II/, https://tox-new.charite.de/protox_II/ accessed on 12 September 2022) tools to predict ADME and toxicity properties, respectively.

### 4.8. Statistical Analysis

Data collected from at least two independent experiments performed in duplicate are expressed as the mean ± SD (standard deviation). They were analyzed using Tukey’s multiple comparison test using GraphPad 8.0 software. Differences were considered statistically significant at *p* < 0.05 (*), *p* < 0.001 (**), and *p* < 0.0001 (***).

## 5. Conclusions

Two promising 2,4-diaminoquinazoline analogs (MMV1578445 (**10**) and MMV1578467 (**7**)) with potent antitrypanosomal activity and high selectivity toward Vero cells emerged from our study. In light of the above evidence, their mechanism of antitrypanosomal action does not include membrane permeabilization or oxidative stress generation. Nevertheless, compound MMV1578445 (**10**) was found to induce *Trypanosoma b. brucei* death through DNA fragmentation as a sign of late apoptosis, while compound MMV1578467 (**7**) showed a moderate ferric iron-reducing ability. Both compounds were predicted to be potent binders of DHFR in silico, eliciting important diaminoquinazoline ring-mediated interactions with key DHFR ligand recognition residues including Val32, Asp54, and Ile160. Further preliminary study in vitro indicated that the inhibitors might be effective DHFR binders. Owing to their favorable pharmacological, physicochemical and predicted ADMET properties, compounds **7** and **10** qualify as suitable starting points for the development of alternative treatments for trypanosomiasis. Hence, we plan to further explore their pharmacological properties and mechanisms of action toward validation as drug candidates.

## Data Availability

All data generated in this study are available in the main manuscript and Appendix A.

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
