# Peer review of "Preliminary Structure–Activity Relationship Study of the MMV Pathogen Box Compound MMV675968 (2,4-Diaminoquinazoline) Unveils Novel Inhibitors of Trypanosoma brucei brucei"

_molecules, 2022, doi:10.3390/molecules27196574_

Round 1
Reviewer 1 Report
The paper authored by Dize et al. deals with the screening of the Pathogen Box against Trypanosoma.
Albeit the search for novel drugs to treat neglected diseases is urgently needed, the novelty of this study is a bit questionable. As the authors correctly cite, the same screening was already performed by others, earlier (Veale et al. and Duffy et al.). However, obviously different hit compounds were identified, hence it might be worth publishing the results.
The title “rudimentary” seems a bit discouraging to me. It would be better to call it “preliminary”, as also stated in the maintext.
Within the introduction it is not clear to me, why DHFR was chosen as the target. The rationale should be introduced already at this point, not just later in the text.
Results part, chapter 2.1. line 85: what is “viz.” ?? not clear to me
Table 1: IC50 for MMV688773. Standard deviation change , to .
Table 3: the information of reported activities against other diseases seems to be irrelevant for me in context of the present study. Could be skipped for more clarity in the table with reduced, more concise information
Chapter 2.2 “rudimentary” à please change like mentioned for the paper title
Line 209: “…inactive at IC50 ≤ 10 μM…” I assume it should be “> 10 µM”
Table 4: Molecular weight is an irrelevant info, please skip for more clarity
Figure 2: is there no standard deviation for the IC50? Just one measurement? Hopefully not…
Figure 3: the docked compounds seems to be protonated at both ring nitrogens. I think this is not possible at physiological pH. Although the predicted pKa are roughly the same, I would assume that the major microspecies is just monoprotonated (the prediction that was made seems to have an isolated view at the single protonation sites). Please doublecheck with freely available software MArvinSketch (ChemAxon). It would be better to discuss the docking pose/binding mode of the true relevant protonation state, which is in my opinion not the double protonated.
Author Response
Reviewer 1
The paper authored by Dize et al. deals with the screening of the Pathogen Box against Trypanosoma.
Albeit the search for novel drugs to treat neglected diseases is urgently needed, the novelty of this study is a bit questionable. As the authors correctly cite, the same screening was already performed by others, earlier (Veale et al. and Duffy et al.). However, obviously different hit compounds were identified, hence it might be worth publishing the results.
The title “rudimentary” seems a bit discouraging to me. It would be better to call it “preliminary”, as also stated in the maintext.
Answer: Thanks for this suggestion. The title has now been revised to read “Preliminary Structure-Activity-Relationship study of the MMV Pathogen Box compound MMV675968 (2,4-diaminoquinazoline) unveils novel inhibitors of Trypanosoma brucei brucei”. Moreover, “rudimentary” was replaced by “preliminary” in the main text.
Within the introduction it is not clear to me, why DHFR was chosen as the target. The rationale should be introduced already at this point, not just later in the text.
Answer: Thank you for raising this important point. We have now revised the introduction to state the rationale for investigating the DHFR enzyme.
Results part, chapter 2.1. line 85: what is “viz.” ?? not clear to me
Answer: “viz.” was deleted and replaced with “including” to make the message clear.
Table 1: IC50 for MMV688773. Standard deviation change , to .
Answer: Thank you. In table 1, the comma (,) was changed to the full stop (.) in the IC50 Tbb standard deviation of MMV688773.
Table 3: the information of reported activities against other diseases seems to be irrelevant for me in context of the present study. Could be skipped for more clarity in the table with reduced, more concise information
Answer: Thank you for this suggestion. The last two columns stating the reported activities against other diseases were deleted.
Chapter 2.2 “rudimentary” à please change like mentioned for the paper title
Answer: The correction was done as suggested, thanks.
Line 209: “…inactive at IC50 ≤ 10 μM…” I assume it should be “> 10 µM”
Answer: This means that at concentrations equal or lower than 10 μM, the compounds were not active.
Table 4: Molecular weight is an irrelevant info, please skip for more clarity
Answer: Thank you for the suggestion. The column showing the molecular weights of compounds has now been deleted from Table 4.
Figure 2: is there no standard deviation for the IC50? Just one measurement? Hopefully not…
Answer: Standard deviation values are now added to the IC50s in Figure 2, thanks for raising this out.
Figure 3: the docked compounds seems to be protonated at both ring nitrogens. I think this is not possible at physiological pH. Although the predicted pKa are roughly the same, I would assume that the major microspecies is just monoprotonated (the prediction that was made seems to have an isolated view at the single protonation sites). Please doublecheck with freely available software MArvinSketch (ChemAxon). It would be better to discuss the docking pose/binding mode of the true relevant protonation state, which is in my opinion not the double protonated.
Answer:
Thank you so much for your keen observation, you are indeed correct about our assumptions around the double protonated form of the compound. After double checking with the proposed software, we have now completely removed the double protonated form from the text and figures and focused our discussions on the three major microspecies: the unprotonated, protonated at position 2, and protonated at position 4 of both compounds.
Reviewer 2 Report
This manuscript entitled Rudimentary Structure-Activity-Relationship study of the MMV Pathogen Box compound MMV675968 (2,4-diaminoquinazoline) unveils inhibitors of Trypanosoma brucei brucei by Boyom et al. described, not for the first time, the screening of the compounds the MMV pathogen box against T. brucei brucei.
As aforementioned, the study was conducted Veale and Hoppe by in 2018 (Med. Chem. Commun., 2018,9, 2037-2044) and the result from this study do not correspond to the previously puclish work. For example, the IC50 of pentamidine from the study and this study show similar IC50s whereas MMV675998's IC50 are 10-fold different. The authors have explained the possible reasons for the discrepancy however the author should show that, under the same experimental setup, MMV675998 can exhibit the antitrypanosomal activity in the level that corresponds to the previous data.
Another issue is that the author use the same compound numbering ID in all tables which makes it difficult for the reader to follow. The experimental results look exciting but it lacks the novelty expected in the publication under the Molecules. The authors should definitely work on a deeper target engagement as the in silico data may not be the best proof that the compounds are targeting at the DHFR of T.brucei brucei.
Therefore, I recommend this manuscript, at this stage, unsuitable for the publication in Molecules.
Author Response
Reviewer 2
This manuscript entitled Rudimentary Structure-Activity-Relationship study of the MMV Pathogen Box compound MMV675968 (2,4-diaminoquinazoline) unveils inhibitors of Trypanosoma brucei brucei by Boyom et al. described, not for the first time, the screening of the compounds the MMV pathogen box against T. brucei brucei.
As aforementioned, the study was conducted Veale and Hoppe by in 2018 (Med. Chem. Commun., 2018,9, 2037-2044) and the result from this study do not correspond to the previously puclish work. For example, the IC50 of pentamidine from the study and this study show similar IC50s whereas MMV675998's IC50 are 10-fold different. The authors have explained the possible reasons for the discrepancy however the author should show that, under the same experimental setup, MMV675998 can exhibit the antitrypanosomal activity in the level that corresponds to the previous data.
Answer: Thank you for pointing out the discrepancies in CI50 values.
Hoping that you are making reference to compound MMV675968, we bear in mind that during drug screening against parasites in culture, differences in assay conditions may result in discrepancies in activity data. Indeed, Veale and Hoppe, (2018) evaluated the antitrypanosomal activity of the Pathogen Box library using an initial parasite density of 2.4x104 parasites/200µL/well with an incubation period of 48 hours, resulting in IC50 MMV675968 = 1.2 µM. On the other hand, Duffy et al. (2017) used an initial parasites density of 1200 cells/mL and incubated for 50 hours, leading to IC50 MMV675998 = 2.07 µM). Meanwhile, in our study, the antitrypanosomal activity of the MMVPBox library was assessed using an initial parasites density of 2x105 cells/mL and incubation period of 72 hours, resulting in IC50 MV675968 = 2.6 µM. The cell density used in the current study was significantly greater than in Veale and Hoppe’s and Duffy et al.’s assays. Indeed, Sykes and Avery, (2009) showed from an antitrypanosomal alamar blue assay that using initial parasite densities of 2,000 cells/mL and 250 cells/mL resulted in IC50 values 4.4 nM and 1.7 nM respectively for pentamidine, suggesting that a reduction in cell inoculum size can result in an increase in parasites sensitivity to compounds (Sykes, M. L.; Avery, V. M. (2009). Development of an Alamar BlueTM Viability Assay in 384-Well Format for High Throughput Whole Cell Screening of Trypanosoma brucei brucei Bloodstream Form Strain 427. American Journal of Tropical Medicine and Hygiene, 81(4), 665–674. doi:10.4269/ajtmh.2009.09-0015).
On another note, using pIC50 (the negative log of the IC50 value when converted to molar) which encourages looking at in vitro assay data logarithmically, rather than IC50 values (encourages linear thinking about an exponential value) is thought to have many advantages. Using pIC50 allows presentation of in vitro data cleanly and in an easy to read form, easy and intuitive average of assay data, and improves how to look at the reliability of in vitro data.
We have now indicated the pIC50 value of each inhibitor in addition to the IC50 values in all result tables of the manuscript. It came out that pIC50 values are within the same range for compound MMV675968 when comparing assay data from the reports of Veale and Hoppe (IC50 1.2 µM /pIC50 5.92), Duffy et al. (IC50 2.07 µM /pIC50 5.68), and this study (IC50 2.6 µM /pIC 5.58).
Another issue is that the author use the same compound numbering ID in all tables which makes it difficult for the reader to follow. The experimental results look exciting but it lacks the novelty expected in the publication under the Molecules. The authors should definitely work on a deeper target engagement as the in silico data may not be the best proof that the compounds are targeting at the DHFR of T.brucei brucei.
Answer: Thanks for this indication. The compound numbering is now removed from tables 2 and 3.
Therefore, I recommend this manuscript, at this stage, unsuitable for the publication in Molecules.
Answer: We have now improved the manuscript as indicated for publication in Molecules.
Reviewer 3 Report
The manuscript entitled “Rudimentary Structure-Activity-Relationship study of the MMV Pathogen Box compound MMV675968 (2,4-diaminoquinazoline) unveils inhibitors of Trypanosoma brucei brucei” by Darline Dize et al., describes the search of an effective antitrypanosomal agent. Authors have identified two promising 2,4-diaminoquinazoline analogs, namely, MMV1578445 and MMV1578467. In general, the manuscript sounds like unfinished research and needs more in-depth studies.
There are several remarks to the manuscript:
- The manuscript has been published as a pre-print in bioRxiv and has DOI: 10.1101/2022.05.20.492762v1.full
- Molecular docking and Dihydrofolate reductase (DHFR) binding properties. The presented results are speculative. Additional in vitro / in vivo experiments are needed.

Author Response
Reviewer 3
The manuscript entitled “Rudimentary Structure-Activity-Relationship study of the MMV Pathogen Box compound MMV675968 (2,4-diaminoquinazoline) unveils inhibitors of Trypanosoma brucei brucei” by Darline Dize et al., describes the search of an effective antitrypanosomal agent. Authors have identified two promising 2,4-diaminoquinazoline analogs, namely, MMV1578445 and MMV1578467. In general, the manuscript sounds like unfinished research and needs more in-depth studies.
There are several remarks to the manuscript:
1. The manuscript has been published as a pre-print in bioRxiv and has DOI: 10.1101/2022.05.20.492762v1.full
Answer: Yes, this manuscript was published as a pre-print in bioRxiv and has DOI: 10.1101/2022.05.20.492762v1.full. This enables wide dissemination of our findings within the scientific community. Of note, the journal, Molecules encourages authors to submit their manuscripts as preprints (https://www.mdpi.com/journal/molecules/instructions#preprints).
2. Molecular docking and Dihydrofolate reductase (DHFR) binding properties. The presented results are speculative. Additional in vitro / in vivo experiments are needed.
Answer: Thank you for this critical suggestion. We have now provided preliminary data pertaining to in vitro inhibitory activity of parasite’s crude protein extract by the parent hit (MMV675968) and the 2 selected analogs (MMV1578445 and MMV1578467).
Reviewer 4 Report
In this manuscript the authors present an in silico-supported experimental SAR study on the analogues of emblematic small molecule MMV Pathogen Box compound MMV675968. Importantly, this research led to identification of two potent antitrypanosomal compounds featuring DHFR- and TR-inhibition with substantially increased selectivity toward Vero cells compared to MMV675968. The presented research is appropriatelly designed and well-conducted, the experiments are described in sufficient details and the interpretation of the results is convincing. Prior to final acceptance of this contribution the authors are expected to cite and briefly present a wider scope of references reporting on studies with MMV675968.
Author Response
Reviewer 4
In this manuscript the authors present an in silico-supported experimental SAR study on the analogues of emblematic small molecule MMV Pathogen Box compound MMV675968. Importantly, this research led to identification of two potent antitrypanosomal compounds featuring DHFR- and TR-inhibition with substantially increased selectivity toward Vero cells compared to MMV675968. The presented research is appropriatelly designed and well-conducted, the experiments are described in sufficient details and the interpretation of the results is convincing. Prior to final acceptance of this contribution the authors are expected to cite and briefly present a wider scope of references reporting on studies with MMV675968.
Answer: Thank you for your kind appreciation of our work. We have further improved the manuscript based on the other reviewers’ comments. Also, references are cited in relation to the scope of investigations conducted on the activity of compound MMV675968, including the antitoxoplasmal, anticandidal, antibacterial, antitrypanosomal, anticryptosporidial and the in vitro inhibition of dihydrofolate reductase (DHFR) enzymes from many pathogens.
Reviewer 5 Report
Manuscript molecules-1858445 "Rudimentary Structure-Activity-Relationship study of the MMV Pathogen Box compound MMV675968 (2,4-diamino-quinazoline) unveils inhibitors of Trypanosoma brucei brucei" by Darline Dize et al. may be accepted for publication in the journal Molecules, subject to the corrections listed below.
1) The manuscript is extensive, and perhaps it would help to track the content if the authors moved some less important parts to Supplementary Information.
2) Correct the sentence in lines 83-85 in the last part "and anti-trypanosomatid drugs, viz.". The meaning of 'viz' must be explained. Also, correct the sentence starting with "Two anti-HAT drugs (pentamidine)" in line 85.
3) Whether the authors considered to use predictors of ADME properties of these chemicals (e.g. SwissADME http://www.swissadme.ch/) or for their toxicities (https://tox-new.charite.de/protox_II/, https ://tox-new.charite.de/protox_II/ ). Please comment or explain if this is not necessary or applicable to your chemical compounds.
4) My suggestion to the authors is to give in the Supplementary file (possible in Excel table) details for 70 compounds mentioned in lines 102-103: "The 70 preselected compounds were submitted to concentration-response study for the determination of their IC50 values .” And, also, indicate those that were used in further analyses.
5) Figure 3 is not mentioned in the text of the manuscript. The same is for data in Table S1 and Table S2. It must be corrected.
Author Response
Reviewer 5
Manuscript molecules-1858445 "Rudimentary Structure-Activity-Relationship study of the MMV Pathogen Box compound MMV675968 (2,4-diamino-quinazoline) unveils inhibitors of Trypanosoma brucei brucei" by Darline Dize et al. may be accepted for publication in the journal Molecules, subject to the corrections listed below.
1) The manuscript is extensive, and perhaps it would help to track the content if the authors moved some less important parts to Supplementary Information.
Answer: Thank you for this comment. However, we believe that the manuscript content fits well as it is, and will enable the reader to grasp the overall information without reading the supplementary information. In addition to the main manuscript, we have provided supplementary materials described the activity data for compounds beyond our current focus.
2) Correct the sentence in lines 83-85 in the last part "and anti-trypanosomatid drugs, viz.". The meaning of 'viz' must be explained. Also, correct the sentence starting with "Two anti-HAT drugs (pentamidine)" in line 85.
Answer: Thank you for pointing out this mistake. “viz.” was deleted and replaced with “including” to make the message clear. The sentence was corrected as appropriate.
3) Whether the authors considered to use predictors of ADME properties of these chemicals (e.g. SwissADME http://www.swissadme.ch/) or for their toxicities (https://tox-new.charite.de/protox_II/, https ://tox-new.charite.de/protox_II/ ). Please comment or explain if this is not necessary or applicable to your chemical compounds.
Answer: Thank you for the suggestion. We have indeed, predicted the ADME properties of the parent hit (MMV675968) and the 2 selected analogs (MMV1578445 and MMV1578467) using the suggested software. It came out that the compounds do not violate the Lipinski’s Ro5. However, they exhibited no BBB permeability, suggesting that further SAR/SPR studies are required to mitigate this particular feature. This is part of our future specific objectives while progressing this hit series. We have provided the ADMET properties prediction data as supplementary material (Table S3).
4) My suggestion to the authors is to give in the Supplementary file (possible in Excel table) details for 70 compounds mentioned in lines 102-103: "The 70 preselected compounds were submitted to concentration-response study for the determination of their IC50 values .” And, also, indicate those that were used in further analyses.
Answer: Thank you for the suggestion. However, sufficient information is already provided in 2 tables submitted as supporting materials.
5) Figure 3 is not mentioned in the text of the manuscript. The same is for data in Table S1 and Table S2. It must be corrected.
Answer: Thank you for the remark. We have now cited Table 3 on page 4, line 154. As well, Table S1 was cited on page 3, line 113 and Table S2 on page 4, line 142 of the revised manuscript with track-changes marked.
Round 2
Reviewer 2 Report
I would like to thank the authors for a prompt and extensive reply. I still do think that the manuscript would be more attractive if the authors can demonstrate a target engagement on the DHFR as the in silico data may not be the best evidence. I still have some minor suggestions on the Fig3 A-D in which the docking poses are difficult to be visualised at they do not show a good depth and the dimension and size of the binding pocket. It might be good to show the mesh surface and even the electronic surface map of the pocket.
Author Response
Response:
Thank you very much for your suggestions on Fig3 A-D. We have now represented the binding pockets in surface formats, with key residues/cofactor shown as sticks to clearly portray the depth and positioning of the ligand within the binding pockets.
Reviewer 3 Report
All my previous remarks were taken into account. Therefore, the presented manuscript can be accepted for publication in the current form.
Author Response
Dear Reviewer,
Thank you very much for taking of your time to review our manuscript.